# MetaOOD: Automatic Selection of OOD Detection Models

**Yuehan Qin[1,*]     Yichi Zhang[1,*]     Yi Nian[2,*]     Xueying Ding[3]     Yue Zhao[1]**
[1]University of Southern California     [2]University of Chicago     [3]Carnegie Mellon University
`{yuehanqi,yzhang42,yzhao010}@usc.edu`,
`nian@uchicago.edu, xding2@andrew.cmu.edu`
*Co-first authors

## Abstract

How can we automatically select an out-of-distribution (OOD) detection model for various underlying tasks? This is crucial for maintaining the reliability of open-world applications by identifying data distribution shifts, particularly in critical domains such as online transactions, autonomous driving, and real-time patient diagnosis. Despite the availability of numerous OOD detection methods, the challenge of selecting an optimal model for diverse tasks remains largely underexplored, especially in scenarios lacking ground truth labels. In this work, we introduce MetaOOD, the first *zero-shot*, *unsupervised* framework that utilizes meta-learning to select an OOD detection model automatically. As a meta-learning approach, MetaOOD leverages historical performance data of existing methods across various benchmark OOD detection datasets, enabling the effective selection of a suitable model for new datasets without the need for labeled data at the test time. To quantify task similarities more accurately, we introduce language model-based embeddings that capture the distinctive OOD characteristics of both datasets and detection models. Through extensive experimentation with 24 unique test dataset pairs to choose from among 11 OOD detection models, we demonstrate that MetaOOD significantly outperforms existing methods and only brings marginal time overhead. Our results, validated by Wilcoxon statistical tests, show that MetaOOD surpasses a diverse group of 11 baselines, including established OOD detectors and advanced unsupervised selection methods.

## 1 Introduction

Out-of-distribution (OOD) detection is the process of identifying data points that deviate significantly from the distribution of the training data. This capability is essential for ensuring the reliability of machine learning models when they encounter new, unseen data (Yang et al., 2021). Common applications of OOD detection include safety-critical systems like autonomous vehicles (Filos et al., 2020; Li et al., 2024a) and medical diagnosis (Ulmer et al., 2020) to prevent erroneous predictions. Notably, many OOD detection algorithms are *unsupervised* due to the high annotation cost, leveraging statistical methods or reconstruction errors from models such as autoencoders to identify deviated samples without labeled OOD data (Yang et al., 2022; Dong et al., 2024; Sun et al., 2022). However, despite the variety of OOD detection algorithms, each adepts at identifying different aspects of in-distribution (ID) and OOD data, there lacks a systematic method for choosing the best OOD detection model(s) under the unsupervised setting *without* labels. It is acknowledged that each OOD detection algorithm might excel in specific scenarios but may not perform well universally due to the no-free-lunch theorem (Wolpert & Macready, 1997). Moreover, without labels, it is difficult to objectively evaluate and compare the performance of different OOD detection models.

**Present Work**. In the most related field, unsupervised outlier detection (OD) has benefitted significantly from meta-learning, with models like MetaOD (Zhao et al., 2021), ELECT (Zhao et al., 2022a), ADGym (Jiang et al., 2024), and HyPer (Ding et al., 2024) demonstrating notable advancements. By leveraging historical performance data, these methods estimate a model's efficacy on new datasets. However, they are not directly adaptable to OOD detection due to several key differences:

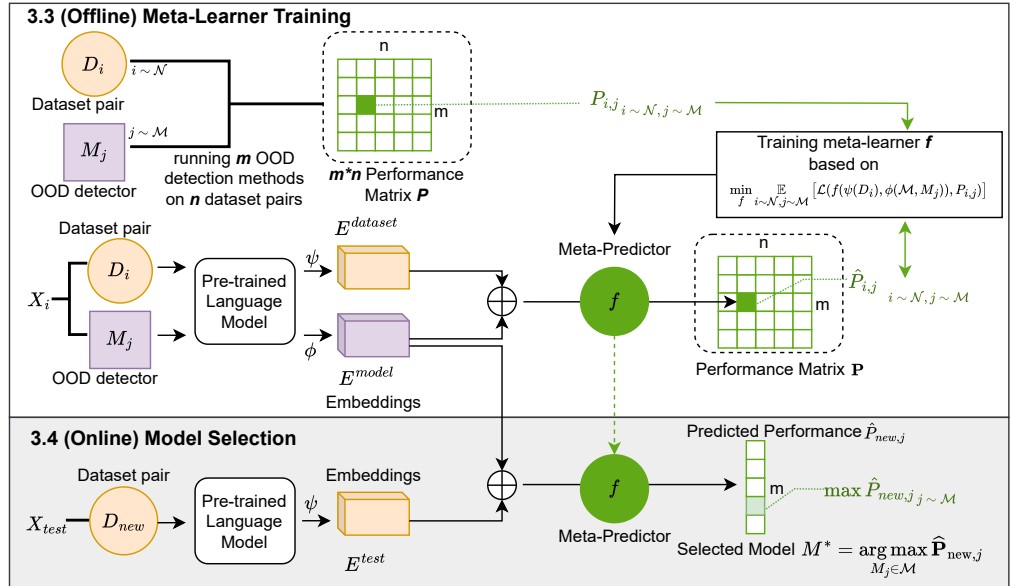

Figure 1: MetaOOD overview (§3.2); offline meta-training phase is shown on the top (§3.3)—the key is to train a meta performance predictor $f$ (denoted in ●) to map language embeddings of the datasets and models to their performance $\mathbf{P}$; the online model selection (§3.4) is shown at the bottom by transferring the meta-predictor $f$ to predict the test data paired with OOD detectors for selection.

*First*, OD and OOD detection differ in their problem settings. OD models are trained on both normal and outlier samples (Zhao et al., 2019; Ma et al., 2023), focusing on anomalies within a single distribution (Zhao et al., 2022b; Zhao, 2024; Chen et al., 2024). In contrast, OOD detection involves training solely on ID data and aims to identify samples from entirely different distributions, often across multiple datasets (Li et al., 2024b; Liu et al., 2024). *Second*, OOD detection tasks primarily deal with images, presenting greater complexity compared to time series and tabular data in OD. *Third*, the embeddings used to measure dataset similarity in OD do not translate well to OOD detection contexts. Current OD embedding generation are largely heuristic and feature-specific (Zhao et al., 2021), inadequate for the diverse and complex nature of OOD datasets, especially when considering the different data modalities. Thus, addressing these challenges with a tailored method for OOD detection model selection is crucial.

**Our Work**. In this study, we introduce MetaOOD, the *first* unsupervised OOD detection model selection method that employs meta-learning. Fig. 1 illustrates the overall workflow of the proposed approach. The idea is that an OOD detector that performed well on similar *historical* datasets is likely to excel on *new* ones. As a meta-learning approach, we train a suite of OOD detection models offline using a variety of carefully curated datasets to gauge their performance across different scenarios during the meta-train phase (Fig.1, top). When a new dataset arrives, we apply knowledge derived from historical data to select an appropriate OOD detection model (Fig. 1, bottom). This selection is based on the similarity between the new dataset and those used in the meta-train phase. To enhance the accuracy of this similarity assessment, we designed two versions of data embeddings via specialized OOD meta-features and language embeddings from language models. As our experiment shows, these language model-based embeddings effectively capture complex, nuanced dataset characteristics that traditional meta-features might miss. We summarize our technical contributions:

- **First OOD Detection Model Selection Framework**. We introduce the first meta-learning-based framework for zero-shot OOD detection model selection *without* training and evaluation.
- **Specialized Embeddings for OOD Datasets and Models**. We use language model-generated features to quantify the similarity among OOD detection tasks, facilitating a better understanding of OOD characteristics and enhancing OOD detection model selection.
- **Effectiveness**. The proposed MetaOOD outperforms eleven well-known model selection methods and unsupervised meta-learners on a testbed with 24 unique test data pairs. It is superior and statistically better than all the baselines w.r.t. average rank, and efficient with small runtime.
- **Accessibility and Reproducibility**. We release the testbed, corresponding code, and the proposed meta-learner at `https://github.com/yqin43/metaood`.

## 2 RELATED WORK

### 2.1 UNSUPERVISED OOD DETECTION MODEL SELECTION

OOD detection faces the challenge of both OOD samples and OOD distributions being unknown during the training phase (Hendrycks et al., 2019; Liang et al., 2018). Consequently, selecting models for OOD detection relies solely on in-distribution or training data (Lee et al., 2018b). Once deployed in an open-world setting, the OOD detector often handles a diverse range of test inputs originating from various OOD distributions (Yang et al., 2021), and consequently, unsupervised methods are expected to be more suitable for selecting OOD detection models (Liu et al., 2020b). Within these unsupervised methods, some recent works focus on actively choosing sample-wise detectors (e.g., choosing different groups of detectors for different input samples) (Xue et al., 2024), while it is not doing actual model selection at the single model level as this work tries to address.

Most existing methods rely on trial-and-error or empirical heuristics. For instance, the simplest approach may be not to actively "select" a model, but just use popular OOD detectors like Maximum Softmax (Hendrycks & Gimpel, 2017) and ODIN (Liang et al., 2017). There are other simple approaches; the confidence level of the ID data may be used as an indicator for OOD detection model selection. However, such a simple method may lead to a low true positive rate (TPR) (Hendrycks & Gimpel, 2017). Another direction is similarity-based methods, where the model selection is based on the similarity or cluster of the dataset, which has also been utilized for algorithm recommendation (Kadioglu et al., 2010; Nikolić et al., 2013; Xu et al., 2012; Misir & Sebag, 2017). We thus include these algorithms as our baselines in §4.

### 2.2 SUPERVISED OOD DETECTION MODEL SELECTION

There has been considerable research into the use of supervised model selection techniques for effectively training and evaluating predictive models on labeled datasets (James et al., 2013; Hastie et al., 2015; Tibshirani, 1996). These methods are particularly valuable in scenarios where generalization to unseen data is critical. Model selection in supervised learning encompasses a variety of strategies, including cross-validation, grid search, and performance metrics such as accuracy and F1 score to assess model efficacy. Randomized (Bergstra & Bengio, 2012), bandit-based (Li et al., 2017), and Bayesian optimization (BO) techniques (Shahriari et al., 2015) are various SOTA approaches to hyperparameter optimization and/or algorithm selection. However, it is notable that such methods do not apply to unsupervised OOD detection model selection, as ground truth values are absent.

### 2.3 REPRESENT DATASETS AND MODELS AS EMBEDDINGS FOR META-LEARNING

In terms of data representation, especially when it comes to meta-learning, embeddings play an important role to measure dataset/task similarity. Traditionally, computational-based meta-features are used as data representation in the meta-learning process (Vanschoren, 2018). More recently, more sophisticated, learning-based meta-features have been designed, as in dataset2vec (Jomaa et al., 2021) and HyPer (Ding et al., 2024). Additionally, there has been a notable integration of language embeddings to encapsulate data features, aiming to enhance model understanding and data representation (Drori et al., 2019; Fang et al., 2024). While the first approach primarily relies on heuristic methods and the latter can be hindered by the slow pace of model training, we adopt language embeddings to represent data for this OOD detection model selection task. In this study, the traditional meta-feature approach is implemented as well for comparison and evaluation.

## 3 METAOOD FOR OOD DETECTION MODEL SELECTION

### 3.1 PRELIMINARIES ON OOD DETECTION

The OOD detection task involves training datasets $\mathbf{X}_{\text{train}}$ sampled from the in-distribution (ID) $\mathcal{P}_{\text{in}}$, denoted $\mathbf{X}_{\text{train}} \sim \mathcal{P}_{\text{in}}$, and testing datasets $\mathbf{X}_{\text{test}}$, which may contain both ID and OOD samples. The goal is to train a model $M$ to classify if a new sample $x \in \mathbf{X}_{\text{test}}$ belongs to $\mathcal{P}_{\text{in}}$.

$$M(x) = \begin{cases} 1 & \text{if } x \in \mathcal{P}_{\text{in}}, \\ 0 & \text{if } x \notin \mathcal{P}_{\text{in}}. \end{cases}$$

Generally, a model is trained exclusively on the ID data to learn to perform tasks such as classification on this dataset. For example, one common approach is using the model's confidence scores as the method $M$ to distinguish between ID and OOD data.

When evaluating an OOD detector on $\mathbf{X}_{\text{test}}$, both ID and OOD test data are present to assess the detector's capability to accurately distinguish between known and unknown data samples. Therefore, in this study, we structure the data into dataset pairs $D = \{\mathbf{X}_{\text{train}}, \mathbf{X}_{\text{test}}\}$, each consisting of the training dataset with ID samples only and the test dataset with both ID and OOD samples.

## 3.2 PROBLEM STATEMENT AND FRAMEWORK OVERVIEW

Given a new, unseen dataset pair for OOD detection, the goal is to select the best model[1] from a heterogeneous set of OOD detection models *without* requiring any model evaluations at test. In this work, we leverage meta-learning to transfer model performance information from prior experiences to the new OOD detection task. Meta-learning (Vanschoren, 2018), often called "learning to learn", is a technique where an algorithm learns from a collection of historical/meta tasks and uses this experience to perform well on new, unseen tasks. The rationale is that an OOD detector is likely to outperform on a new dataset if it excels in a similar historical dataset. This approach is particularly useful when evaluation is infeasible or costly due to a lack of labels or the need for rapid adaptation.

The proposed meta-learner, MetaOOD, relies on:

- A collection of $n$ historical (i.e., meta-train) OOD detection dataset pairs, $\mathcal{D}_{\text{train}} = \{D_1, \ldots, D_n\}$ with ground truth labels, i.e., $D = \{\mathbf{X}_{\text{train}}, (\mathbf{X}_{\text{test}}, \mathbf{y}_{\text{test}})\}$.
- Historical performance $\mathbf{P}$ of the pre-set model set $\mathcal{M} = \{M_1, \ldots, M_m\}$ (with $m$ models), on the meta-train datasets. We refer to $\mathbf{P} \in \mathbb{R}^{n \times m}$ as the performance matrix, where $\mathbf{P}_{i,j}$ corresponds to the $j$-th model $M_j$'s performance on the $i$-th meta-train dataset pair $D_i$.

Our objective is to choose the best candidate OOD detection model $M \in \mathcal{M}$, given a new pair of datasets $D_{\text{new}} = \{\mathbf{X}_{\text{train}}^{\text{new}}, \mathbf{X}_{\text{test}}^{\text{new}}\}$ as input, where we have no ground truth labels $\mathbf{y}_{\text{test}}^{\text{new}}$ for evaluation.

**Problem 1 (OOD detection model selection)** *Given a new input dataset $D_{new} = \{\mathbf{X}_{train}^{new}, \mathbf{X}_{test}^{new}\}$ (detect OOD samples on $\mathbf{X}_{test}^{new}$ with only in-distribution data $\mathbf{X}_{train}^{new}$ and no labels), select a model $M \in \mathcal{M}$ to employ on the new (test) task.*

The problem is similar to the OD model selection task such as MetaOD, ELECT, and HPOD (Zhao, 2024). However, in contrast to OD model selection, which focuses solely on a single dataset, OOD detection model selection requires the consideration of both training and test datasets. This necessity stems from the need to take into account the similarities and differences among both datasets that impact pre-training on the ID data and the actual OOD detection on the test data, all crucial to measuring the inherent characteristics of OOD detection task.

In a nutshell, our MetaOOD consists of two phases: (*i*) offline (meta-) training of the meta-learner on $\mathcal{D}_{\text{train}}$ where the goal is to learn the mapping from OOD detection models' performance on various (historical/meta) datasets, and (*ii*) online model selection that uses the meta-train information to choose the model at test time for the new dataset $D_{\text{new}}$. Fig.1 outlines the workflow and key elements of MetaOOD, with the offline training phase shown in white and the online model selection stage shown in grey. The details of the two phases are discussed in §3.3 and §3.4, respectively.

## 3.3 OFFLINE META-TRAINING

During offline training, we generate embeddings for dataset pair $\mathcal{D}_{\text{train}}$ and method $\mathcal{M}$, and train the latent mapping from these embeddings to the performances $\mathbf{P}$. The meta-learner can generalize and select the best-performing model for new, unseen datasets by learning the relationship between $\{\mathcal{D}_{\text{train}}, \mathcal{M}\} \to \mathbf{P}$. Note this training process is supervised. Prior works have shown that performance mapping is empirically learnable, although imperfect, in related fields like OD (Zhao, 2024).

To predict the performance of the *candidate model* on *a new dataset pair*, we propose training a meta-predictor as a regression problem. The input to the meta-predictor consists of $E_i^{\text{meta}}, E_j^{\text{model}}$, corresponding to the embedding of the $i$-th dataset pair and the embedding of the $j$-th OOD detector. Dataset embedding of dataset pair $\mathcal{D}$ is denoted as $E^{\text{data}} = \psi(\mathcal{D})$, and method embedding extracted from $\mathcal{M}$ is denoted as $E^{\text{model}} = \phi(\mathcal{M}, M)$; we provide more details below on the embedding generation in §3.3.1. Our goal is to train the meta-predictor $f$[2] to map the characteristics of the datasets and the OOD detectors to their corresponding performance ranking across all historical dataset pairs. The steps of the (offline) meta-train are shown in Fig. 1, top and Appx. Algo. 1.

$$f : E_i^{\text{data}}, E_j^{\text{model}} \to \mathbf{P}_{i,j}, \quad i \in \{1, \ldots, n\}, \quad j \in \{1, \ldots, m\} \tag{1}$$

---

[1]The use of the term 'models' is more for following the tradition of model selection research.

[2]The format of $f$ can be any regression models; in this work, we use an XGBoost (Chen & Guestrin, 2016) model due to its balance of simplicity and expressiveness, as well as strong feature selection characteristic.

### 3.3.1 DATA AND MODEL EMBEDDINGS

Data and model embeddings, as inputs of $f$, offer compact, standardized context about the data and models utilized in the meta-learning process. Instead of directly using data with arbitrary sizes, we hope to generate embeddings that can effectively represent the data and help develop models that can adapt rapidly to new tasks. In this work, we try two types of data and model embeddings: (1) *classical* (data) meta-features and (method) one-hot encoding embedding, and (2) *language model-based approach* to get data and model embeddings via language models.

**Classical Way: Data Embedding via Meta-Features and Model Embeddings via One-hot Encoding**. Meta-features, or "features of features", are attributes used in meta-learning, model selection, and feature engineering to provide higher-level information about a dataset or its characteristics (Vanschoren, 2018). These include basic statistics, such as the mean and maximum values of the data, as well as outputs or performance metrics from preliminary models tested on the dataset. Meta-features facilitate understanding the nuances of various learning tasks and guide the selection or configuration of models for new tasks by drawing parallels with previously encountered tasks. In this work, we first create meta-features that capture OOD characteristics across different dataset pairs. These features aim to identify similarities between data in the meta-training database and test data, enhancing the model selection process based on prior successful applications. To achieve this, we categorize the extracted meta-features into ***statistical*** and ***landmarker*** categories.

*Statistical meta-features* capture the underlying data distributions' characteristics, including variance, skewness, and other relevant statistics of individual features and their combinations. For image datasets used in OOD detection, we incorporate meta-features that reflect pertinent image characteristics (Aguiar et al., 2019), such as: (*i*) Color-based: Simple statistical measures from color channels; (*ii*) Border-based: Statistical measures obtained after applying border-detector filters; (*iii*) Histogram: Statistics from histograms of color and intensity; and (*iv*) Texture features: Values derived from an image's texture, analyzed using the co-occurrence matrix and Fast Fourier Transform.

*Landmarker features* summarize the performance of specific learning algorithms on a dataset, providing quick and effective estimates of algorithm performance. These features, derived from evaluating simple/fast models, offer a snapshot of dataset characteristics and approximate the performance of more sophisticated models in specific tasks. For OOD detection, our landmarkers focus on features related to Softmax probability outputs that can be rapidly generated without model fitting. Detailed descriptions and a complete list of our OOD meta-features are in Appx. Tab. B.

**New Approach: Data Embedding via Language Models**. Meta-features, often handcrafted and heuristic, have limitations in scalability and adaptability. This manual approach to feature engineering can be labor-intensive and may not capture all the nuanced relationships within OOD data, potentially affecting the efficacy of model selection. In contrast, leveraging language models offers a transformative approach for generating data embeddings (Peng et al., 2024). These models have been recently utilized to generate embeddings from textual descriptions of datasets and methods, capturing essential information that reflects the datasets' and methods' intrinsic properties due to their comprehensive training (Drori et al., 2019). Given that language models are designed to process text inputs, we hypothesize that utilizing text descriptions of data as input to a language model for embedding generation could effectively encapsulate the inherent features of the data. In this study, we experiment with various language models to produce the language embeddings.

To capture the dataset features, we input dataset metadata such as size, object type, and description. For example, the input for the CIFAR-10 dataset (Krizhevsky, 2009) is formatted as:

```
Contains images of 10 types of objects, including airplanes, cars,
birds, cats, deer, dogs, frogs, horses, ships, and trucks.  Each
image is a small 32x32 RGB image.
```

For OOD detection model embeddings, we take the textual descriptions of methods from the Pytorch-OOD library (Kirchheim et al., 2022) as input. This includes detailed information about the OOD detector, such as its components, whether it needs fitting, and what is used as OOD indicator. An example of this is the description for the Openmax method (Bendale & Boult, 2016):

```
Determines a center for each class in the logits space of a
model, and then creates a statistical model of the distances of
correctly classified inputs.  It uses extreme value theory to
```

```
detect outliers by fitting a Weibull function to the tail of the
distance distribution.  The activation of the unknown class is
used as the outlier score.
```

The complete lists of the datasets and methods used are listed in §4.1. In brief, this approach leverages language models' natural language processing capabilities to transform qualitative descriptions into quantitative embeddings. Compared to the traditional meta-feature approach, it is more computationally efficient and has better generalization ability. Also, it is interesting to see whether we could use language models to choose an OOD detector directly. We provide all these analyses in §4.

## 3.4 ONLINE MODEL SELECTION

In the online model selection process, we generate the embeddings for the test dataset pair $D_{\text{new}}$ and reuse the model embeddings of $\mathcal{M}$, apply the trained meta performance predictor $f$ from the offline stage to predict different OOD detection models' performance, and select the model with the highest predicted performance, as described in Eq. (2).

$$M^* := \underset{M_j \in \mathcal{M}}{\arg\max} \widehat{\mathbf{P}}_{\text{new},j}, \quad \text{where} \quad \widehat{\mathbf{P}}_{\text{new},j} = f(E_{\text{new}}^{\text{meta}}, E_j^{\text{model}}) \tag{2}$$

Specifically, for a new dataset pair, we acquire the predicted relative performance ranking of different OOD detection methods using the trained $f$, and select the top-1 method[1], as shown in Eq. (2). It is important to note that this procedure is *zero-shot*, and does not require any training on the test sample. The (online) model selection steps are given in Fig. 1, bottom and Appx. Algo. 2.

## 4 EXPERIMENTS

Our experiments answer the following research questions (RQ): **RQ1** (§4.3): How effective is the proposed MetaOOD in unsupervised OOD detection model selection in comparison to other leading baselines? **RQ2** (§4.4.1): How do different design choices in MetaOOD impact its effectiveness? **RQ3** (§4.4.2): How much time overhead/saving MetaOOD brings to OOD detection in general?

## 4.1 EXPERIMENT SETTING

**The model Set** $\mathcal{M}$. We compose a model set $\mathcal{M}$ to choose from with 11 popular OOD detection models as shown in Tab. 1, covering different types of detection methods.

**The OOD Datasets**. We utilize the train-test split of datasets preprocessed as described in (Yang et al., 2022). To summarize, we create our ID-OOD dataset pairs using the following datasets:

1. ID Datasets: CIFAR10 (Krizhevsky, 2009), CIFAR100 (Krizhevsky, 2009), ImageNet (Deng et al., 2009), FashionMNIST (Xiao et al., 2017)

2. Classic OOD Group: CIFAR10, CIFAR100, MNIST (Deng, 2012), Places365 (Zhou et al., 2018), SVHN (Netzer et al., 2011), Textures (Cimpoi et al., 2014), TIN (Le & Yang, 2015)

3. Large-Scale OOD Group: SSB_hard (Vaze et al., 2022), NINCO (Bitterwolf et al., 2023), iNaturalist (Horn et al., 2017), Textures (Cimpoi et al., 2014), OpenImage-O (Wang et al., 2022)

where semantic overlap images between the ID data and the OOD data are removed (1,203 images from TIN, and 1,305 images removed from MNIST, SVHN, Textures, and Places365).

We construct the ID-OOD dataset pair, and set the training and testing set as follows: (i) **Training:** CIFAR10 from ID and OOD from the classic OOD group shown above; and (ii) **Testing:** CIFAR100, ImageNet, and Fashion-MNIST from ID, and OOD from large-scale OOD dataset group. The other ID datasets undergo a similar train test split, which results in 24 unique testing pairs. Our split guarantees that no dataset in the testing pair has been seen/leaked during the meta-train process.

| Category | OOD Detection Model |
|---|---|
| Probability Based | MSP (Hendrycks & Gimpel, 2017) |
| | MCD (Gal & Ghahramani, 2016) |
| | KLM (Hendrycks et al., 2022) |
| | Entropy (Chan et al., 2020) |
| Logit-based | MaxLogit (Hendrycks et al., 2022) |
| | Openmax (Bendale & Boult, 2016) |
| | EnergyBased (Liu et al., 2020a) |
| | ODIN (Liang et al., 2017) |
| Feature-based | Mahalanobis (Lee et al., 2018a) |
| | ViM (Wang et al., 2022) |
| | $k$NN (Cover & Hart, 1967) |

Table 1: OOD detection models in this study.

---

[1]It may choose the top-$k$ candidates as an ensemble, while this work focuses on top-1 selection.

**Hardware**. For consistency, all models are built using the pytorch-ood library (Kirchheim et al., 2022) on NVIDIA RTX 6000 Ada, 48 GB RAM workstations.

**Training the meta-predictor** $f$ (see details in §3.3). In this work, we use an XGBoost (Chen & Guestrin, 2016) model as $f$ due to its simplicity and expressiveness. Meanwhile, we use the data and model embeddings generated by the pre-trained BERT-based all-mpnet-base-v2 model by Hugging Face (Reimers & Gurevych, 2019). We provide more ablations on $f$ in §4.4.1.

**Evaluation**. To compare the performance of MetaOOD and baselines, we examine the performance rank[1] of the OOD detector chosen by each method among all the candidate detectors through a boxplot and the rank diagram (which is the average across all dataset pairs). Clearly, the best rank is 1, and the worst is 12 (i.e., 11 baselines and MetaOOD). To compare our algorithm with a baseline, we employ the pairwise Wilcoxon rank test on performances across dataset pairs (significance level $p < 0.05$). The full selection results are in Appx. Tab. C.

## 4.2 Model Selection Baselines

We select the baselines following the literature in meta-learning for unsupervised model selection (Zhao et al., 2021; 2022a; Jiang et al., 2024; Park et al., 2023) with four categories:

*(a) No model selection or random selection*: always employs either the ensemble of all the models or the same single model, or randomly selects a model: **(1) Maximum Softmax Probability (MSP)** (Hendrycks & Gimpel, 2017), as a popular OOD detection model, uses the maximum softmax score of a neural network's output as threshold to identify whether an input belongs to the distribution the network was trained on. **(2) ODIN** (Liang et al., 2017) applies temperature scaling and small perturbations to the input data, which helps to amplify the difference in softmax scores between ID and OOD samples. **(3) Mega Ensemble (ME)** averages OOD scores from all the models for a given dataset. As such, ME does not perform model selection but rather uses *all* the models. **(4) Random Selection (Random)** randomly selects a model from the pool of candidate models.

*(b) Simple meta-learners* that do not involve optimization: **(5) Global Best (GB)** is the *simplest meta-learner* that selects the model with the largest average performance across all meta-train datasets. GB does *not* use any meta-features. **(6) ISAC** (Kadioglu et al., 2010) clusters the meta-train datasets based on meta-features. Given a new dataset pair, it identifies its closest cluster and selects the best model of the cluster. **(7) ARGOSMART (AS)** (Nikolić et al., 2013) finds the closest meta-train dataset (1 nearest neighbor) to a given test dataset, based on meta-feature similarity, and selects the model with the best performance on the 1NN dataset.

*(c) Optimization-based meta-learners* which involves a learning process: **(9) ALORS** (Misir & Sebag, 2017) factorizes the perf. matrix to extract latent factors and estimate perf. as the dot product of the latent factors. A regressor maps meta-features onto latent factors. **(9) NCF** (He et al., 2017) replaces the dot product used in ALORS with a more general neural architecture that predicts performance by combining the linearity of matrix factorization and non-linearity of deep neural networks. **(10) MetaOOD_0** uses manually crafted heuristic meta-features, which consist of both statistical and landmarker features for dataset meta-features, along with one-hot encoding for model embeddings, as discussed in §3.3.1.

*(d) Large language models (LLMs) as a model selector*: **(11) GPT-4o mini** (OpenAI et al., 2024) is used as a zero-shot meta-selector. The dataset and method descriptions are directly provided to the LLM, allowing it to select the methods based on these descriptions. Note there is no meta-learning here. The details are presented in Appx. §B.1.

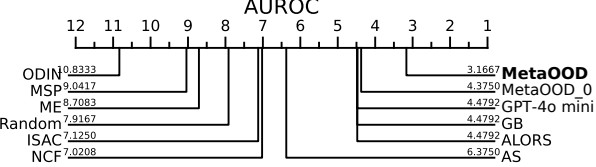

Figure 2: Average rank (lower is better) of methods w.r.t. performance across datasets; MetaOOD outperforms all baselines with the lowest rank.

---

[1]In this work, we choose Area Under the Receiver Operating Characteristic Curve (AUC-ROC), or ROC, as the performance metric; can replace by any other metrics at interest.

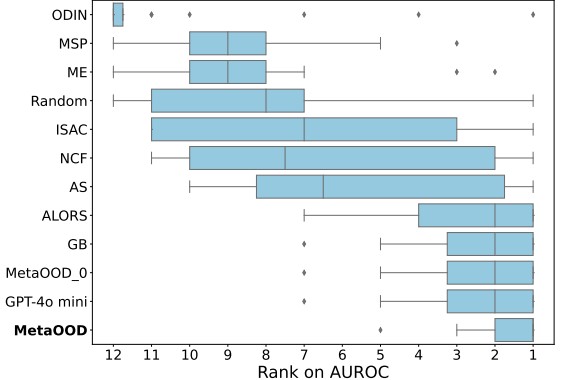

Figure 3: Boxplot of the rank distribution of MetaOOD and baselines (the lower, the better). MetaOOD is the lowest/best.

| Ours | Baseline | p-value |
|---|---|---|
| **MetaOOD** | MetaOOD_0 | 0.0357 |
| **MetaOOD** | ME | <0.0001 |
| **MetaOOD** | AS | 0.0004 |
| **MetaOOD** | ISAC | 0.0001 |
| **MetaOOD** | ALORS | 0.018 |
| **MetaOOD** | Random | 0.0005 |
| **MetaOOD** | MSP | <0.0001 |
| **MetaOOD** | ODIN | <0.0001 |
| **MetaOOD** | NCF | 0.0027 |
| **MetaOOD** | GB | 0.018 |
| **MetaOOD** | GPT-4o mini | 0.018 |

Table 2: Wilcoxon signed-rank test result. MetaOOD is statistically better than all the baselines.

### 4.3 Overall Results

First, we present the rank distribution of the actual rank of the top-1 OOD detector chosen by each model selection method across the 24 test data pairs in Fig. 3. To compare two model selection algorithms (e.g., ours with a baseline), we perform Wilcoxon rank test on the AUROC value of the top-1 models selected by our method and the baseline method, as shown in Tab. 2. Also, we show the aggregated average rank plot in Fig. 2, where the average performance rank of the OOD detection model selected by each algorithm is shown. Here are the **key findings** of the results:

**1. MetaOOD outperforms all baselines**. As shown in Fig. 3, MetaOOD consistently performs well with small variance. On the aggregated level, the average rank of MetaOOD surpasses all of the 11 baselines in Fig. 2. Moreover, MetaOOD has demonstrated statistically superior performance compared to all these baselines, as shown in Tab. 2. Such enhancement in performance is indicative of MetaOOD's robust approach to tackling complex datasets while generating stable performance. We credit its superiority to the combination of a meta-learning framework and the use of language models in embedding datasets and models.

**2. Optimization-based meta-learning methods** (i.e., MetaOOD, MetaOOD_0 and ALORS) perform better in unsupervised model selection over other baselines. One reason is that meta-learning leverages prior experiences to adapt to new tasks. It optimizes the learning process by extracting common patterns and representations across different tasks, enhancing the model's generalization ability. Meanwhile, the optimization process helps models converge to an optimal solution efficiently. Compared to simple meta-learners, such optimization-based methods take advantage of the meta-features for a better mapping of the model performance. For instance, AS and ISAC do not involve optimization and resemble finding the closest sample or closest sample cluster for choosing the top-1 detector. They are inferior to MetaOOD as there is no optimization on the OOD detector selection, but fully depends on the data embeddings for performance knowledge transfer.

**3. The underperformance of no model selection and random selection baselines justifies the need for OOD model selection**. We analyze their performance below.

- **ME**: Averaging the OOD detection scores of all models does not yield strong results, as demonstrated in Fig. 2 and 3. This may stem from certain models consistently underperform across various datasets, and combining all models indiscriminately reduces overall effectiveness. While using selective ensembles might offer improvements (Zhao & Hryniewicki, 2019), constructing ensembles from numerous models can be impractical due to high costs. In contrast, MetaOOD learns to make optimal selections without constructing any models, allowing it to operate efficiently during testing.
- **Random**: According to Fig. 3 and Tab. 2, random selection performs worse than all the meta-learners. This indicates that all the meta-learner baselines we chose do have some improvements compared to random choice, and it is not advised to select an OOD detection model randomly.
- **ODIN and MSP**: As expected, a single method does not perform well across all datasets. This result is not surprising since different OOD detectors emphasize various aspects of the datasets, and real-world datasets have diverse characteristics. Relying on a single approach tends to limit the scope of solutions, making it difficult to capture the distribution shift among different datasets.

**4. LLM acts as a reasonable zero-shot model selector in OOD detection model selection**. This highlights the potential of LLM in model selection, while it can be improved by meta-learning. While the performance of LLM as a selector may be slightly worse (ranked as top 3) than our MetaOOD, LLM consistently selects the globally best model (Openmax (Bendale & Boult, 2016)) across diverse tasks, with *no prior knowledge* about the performances of different OOD detection methods on different models. This is valuable when prior knowledge is limited or when facing novel tasks, highlighting the potential of LLMs in unsupervised model selection. It can be improved via meta-learning by receiving more performance-related information. Arguably, MetaOOD leverages language embeddings along with the meta-learning, thus achieving better performance. Future work may consider LLMs' robustness and trustworthiness in model selection Huang et al. (2025).

## 4.4 ABLATION STUDIES AND ADDITIONAL ANALYSES

### 4.4.1 ABLATION STUDY

In §3.3.1, we discussed how to train the meta-performance predictor $f$, with the choices on *data and model embeddings* and $f$ itself. Here, we conduct two ablations on these choices:

**Ablation 1 on Data and Model Embeddings**. We compare the MetaOOD's performance on different LLM-generated embeddings and different combinations of embeddings. The variants include:

- **OpenAI text-embedding-3-small (OpenAI_emb) (Reimers & Gurevych, 2019)**: It is an embedding model designed by OpenAI to generate compact and meaningful representations of text for various natural language processing tasks.
- **LLama2-7b (Llama) (Touvron et al., 2023)**: LLama2 is an advanced language model that provides enhanced capabilities in a variety of applications, including text summarization, translation, and conversational AI. Since it is a transformer style decoder-based model, the sequence level embeddings are produced by pooling token level embeddings together[1].
- **MetaOOD_0** uses meta-features for dataset embeddings, along with the simple one-hot encoding for model embeddings (see §3.3 for more details).

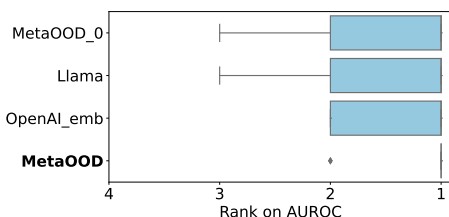

Figure 4: Ablation study on different data and model embeddings. MetaOOD has better performance over its variants.

Figure 5: Ablation study on different choices of meta-predictor $f$. Tree-based models have better performance.

Fig. 4 presents ablation studies results on MetaOOD that uses different embedding variants with $f$ fixed to XGBoost. The results indicate that language models are highly effective at generating embeddings. MetaOOD, along with the other two language model-generated embeddings, demonstrate solid performance. We observed that the BERT-based embeddings perform slightly better than those derived from LLMs. Fig. 6 shows the dataset embeddings produced by different language models, with the embeddings reduced to 2D using t-SNE. Note that MetaOOD (BERT-based embedding) is slightly better to capture similar datasets (e.g. Texture and Textures) than decoder-based model embeddings. This could be attributed to the decoder model's causal attention, where the representation of a token is influenced only by preceding tokens, making it less effective for text embedding tasks as it restricts capturing information from the entire input sequence (BehnamGhader et al., 2024).

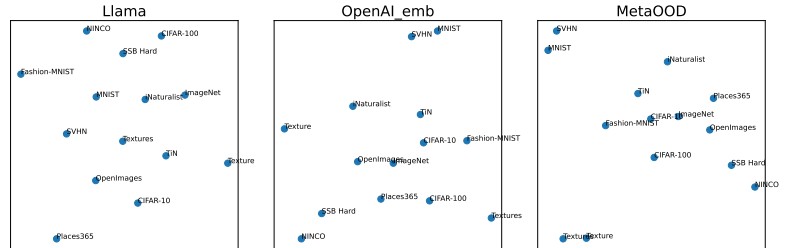

Figure 6: Visualization of dataset embeddings generated by different language models. Bert-based model performs slightly better (e.g., Texture and Textures datasets are close).

---

[1]Usually this is done by averaging the token level embeddings or using the last token. In this study, we opt for the latter approach.

**Ablation 2 on the Choice of Meta-predictor** $f$. We evaluate the performances of **MetaOOD_NN** and **OpenAI_emb_NN**, which use the language embeddings to train a *two-layer MLP meta-predictor*. The result is shown in Fig. 5. Consistent with the finding in (Jiang et al., 2024), using a tree-based model such as *XGBoost* for meta-predictor leads to more stable results and enhanced performance compared to initializing with neural network models. Additionally, tree-based models offer superior feature selection capabilities and greater interpretability. In contrast, neural networks, while powerful, can be more sensitive to initialization and hyperparameter choices, leading to less predictable performance. This finding underscores the significance of selecting features of importance during meta-learning, especially in complex tasks where stability and performance are critical.

### 4.4.2 Runtime Analysis of MetaOOD

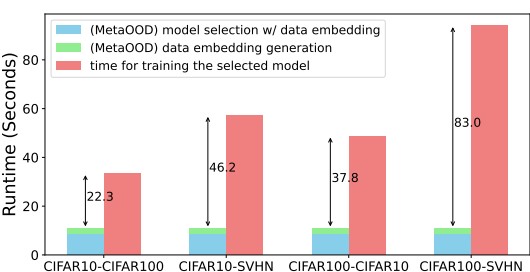

Figure 7: Runtime of MetaOOD vs. selected OOD detector. MetaOOD incurs a small overhead (difference shown with black arrows).

One of the big advantages of MetaOOD over MetaOOD_0 is the fast dataset embedding generation via language models, making the model selection overhead negligible to the actual OOD detection fitting on the image datasets. We demonstrate the time of MetaOOD and the fitting of the selected OOD detector on selected dataset pairs in Fig. 7. Notably, the language embedding for both datasets and models and online model selection via $f$ just takes seconds to finish. Thus, the MetaOOD only brings the marginal cost to the entire OOD detection pipeline while finding a top-performing model.

## 5 Conclusion, Limitations, and Future Directions

In this work, we introduced MetaOOD, the *first* unsupervised out-of-distribution (OOD) detection model selection framework. This meta-learner utilizes an extensive pool of historical data on OOD detection models and dataset pairs, employing language model-based embeddings to enhance the model selection process based on past performances. Despite its innovative approach, MetaOOD depends on the availability and quality of historical dataset pairs. This reliance may limit its effectiveness in scenarios where such data are sparse or less similar. Moreover, the current framework is designed for single-modality data, which restricts its application in highly diverse or multimodal environments. Looking ahead, we plan to broaden our testbed to include a more diverse group of datasets and models, thereby enhancing the meta-learning capabilities of MetaOOD. We also aim to extend MetaOOD to support top-$k$ selection, offering a range of viable models rather than a single recommendation. Additionally, equipping MetaOOD with an uncertainty quantification mechanism will enable it to output an "I do not know" response when applicable, further refining its utility in complex scenarios where no suitable meta-train knowledge can be transferred.

## Broader Impact Statement, Ethics Statement, and Reproducibility

**Broader Impact Statement**: MetaOOD revolutionizes OOD detection model selection by enabling practitioners to choose appropriate models for unlabeled tasks automatically. This is particularly crucial in sectors like healthcare, finance, and security, where rapid adaptation to new data types can significantly enhance system reliability and prevent critical errors. By providing a systematic approach to select the most effective models, MetaOOD promotes robust applications in dynamically changing environments, ensuring ongoing reliability and accuracy in critical systems.

**Ethics Statement**: Our research adheres to the ICLR Code of Ethics, ensuring that MetaOOD is developed and applied with ethical considerations at the forefront, particularly in privacy, bias, and fairness across diverse applications. By facilitating more accurate and unbiased model selection, MetaOOD helps mitigate potential ethical risks in its deployments, such as in surveillance and healthcare, promoting fairness and protecting privacy. Continuous ethical evaluations accompany MetaOOD's development to ensure it meets societal and legal standards.

**Reproducibility Statement**: We advocate reproducibility in MetaOOD. Comprehensive documentation of our methodologies and experimental designs is detailed in the main text and appendices. We have made our code, testbed, and meta-learner fully accessible at `https://github.com/yqin43/metaood`, providing the resources for replication and exploration.

## ACKNOWLEDGMENTS

This work was partially supported by the National Science Foundation under Award No. 2346158. Any opinions, findings, and conclusions or recommendations expressed in this material are those of the authors and do not necessarily reflect the views of the National Science Foundation.

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

## SUPPLEMENTARY MATERIAL FOR METAOOD

## A    DETAILS ON METAOOD

### A.1    PSEUDO-CODE FOR META-TRAIN AND ONLINE MODEL SELECTION

We discussed meta-training and online model selection in §3.3 and §3.4, respectively. Here is the pseudo-code for the two phases.

---

**Algorithm 1** Offline OOD detection meta-learner training

---

**Input:** Meta-train database $\mathcal{D}_{\text{train}}$, model set $\mathcal{M}$
**Output:** Meta-learner $f$ for OOD detection model selection
1:  Train and evaluate $\mathcal{M}$ on $\mathcal{D}_{\text{train}}$ to get performance matrix $\mathbf{P}$
2:  **for** $i \in \{1, \dots, n\}$ **do**
3:      Extract data embedding $E_i^{\text{meta}} = \psi(D_i)$
4:      **for** $j \in \{1, \dots, m\}$ **do**
5:          Encode methods set as $E_j^{\text{model}} = \phi(\mathcal{M}, M_j)$
6:          Train $f$ by Eq. (1) with the $j$-th model on the $i$-th dataset
7:      **end for**
8:  **end for**
9:  **return** the meta-learner $f$

---

**Algorithm 2** Online OOD detection model selection

---

**Input:** the meta-learner $f$, New ID-OOD dataset pair $D_{\text{new}}$
**Output:** Selected model for $D_{\text{new}}$
1:  Extract data embedding, $E_{\text{test}}^{\text{data}} := \psi(D_{\text{new}})$
2:  **for** $j \in \{1, \dots, m\}$ (for clarity, written as a for loop) **do**
3:      Encode methods set as $E_j^{\text{model}} = \phi(\mathcal{M}, M_j)$
4:      Predict the $j$-th model performance by the meta-learner $f$, i.e., $\widehat{\mathbf{P}}_{\text{new},j} := f(E_{\text{new}}^{\text{data}}, E_j^{\text{model}})$
5:  **end for**
6:  **return** the model with the highest predicted perf. by Eq. (2)

---

### A.2    DETAILS ON NOTATIONS

The following notations are used in Fig. 1, which provides a comprehensive MetaOOD overview (§3.2).

| Notations | Description |
|---|---|
| $\mathcal{L}$ | Training Loss |
| $\mathcal{M}$ | # OOD Detection Methods |
| $\mathcal{N}$ | # Dataset Pairs |
| $\phi$ | Embedding Notation for OOD Detection Methods |
| $\psi$ | Embedding Notation for Dataset Pairs |
| $P_{i,j}$ | Performance of OOD Detection Method $j$ on Dataset Pair $i$ |
| $\hat{P}_{i,j}$ | Predicted Performance of OOD Detection Method $j$ on Dataset Pair $i$ |
| $f$ | Meta-predictor |
| $X$ | Input Sample |

Table A: Notations with details used in Fig. 1

### A.3    DETAILS ON META-FEATURES

The classical way of data embedding via meta-features is discussed in §3.3.1. Tab. B presents the complete list of meta-features constructed in this study.

| Category | Description | Variants |
|---|---|---|
| **Statistical Features** | | |
| Sample features | $\mu, \tilde{X}, \sigma^2, \min_X, \max_X, \sigma$ | mean, median, var, min, max, std |
| | percentile $P_i$ | q1, q25, q75, q99 |
| | $q75 - q25$ | IQR |
| | $\frac{\mu}{\max_X}, \frac{\tilde{X}}{\max_X}$ | normalized mean, normalized median |
| | $\max_X - \min_X, \text{Gini}(X)$ | sample range, sample gini |
| | $\text{median}(X - \tilde{X})$ | median absolute deviation |
| | $\text{avg}(X - \tilde{X})$ | average absolute deviation |
| | $\frac{q75-q25}{q75+q25}$ | Quantile Coefficient Dispersion |
| | | Coefficient of variance |
| | If a sample differs from a normal dist. | normality |
| | | 5th to 10th moments |
| | skewness, $\frac{\mu^4}{\sigma^4}$ | skewness, kurtosis |
| | $\mathcal{M}$ | Co-occurence Matrix |
| Image features | color-based | mean, std of the HSV channel |
| | color-based | std of the intensity channel |
| | color-based | entropy of the RGB channel |
| | histogram | std of the (RGB, HSV, intensity) channel |
| | border | Average white pixels |
| | border | Average Hu Moments of sobel image |
| | texture | |
| | (Co-occurence Matrix) | contrast mean, std |
| | (Co-occurence Matrix) | dissimilarity mean, std |
| | (Co-occurence Matrix) | homogeneity mean, std |
| | (Co-occurence Matrix) | energy mean, std |
| | (Co-occurence Matrix) | correlation mean, std |
| | (Co-occurence Matrix) | entropy mean, std |
| | (FFT) | entropy mean, std |
| | (FFT) | inertia mean, std |
| | (FFT) | energy mean, std |
| | (FFT) | homogeneity mean, std |
| Dataset features | $n, p$ | num of samples, num of features |
| | $d, c$ | dim, num of class |
| | EMD | Earth Mover's Distance |
| **Landmarker Features** | | |
| Probability-based | softmax probability | mean, std, min, max |
| | | entropy, range |
| | | top1 softmax probability |
| | | top2 softmax probability |
| | | considence margin |
| | | skewness, kurtosis |

Table B: Selected MetaOOD features. Part of the statistical features are based on Vanschoren (2018); Zhao et al. (2021), and specialized landmarker features are newly designed for OOD detection. See §3.3 for more details.

# B ADDITIONAL EXPERIMENT SETTINGS AND RESULTS

## B.1 PROMPTS TO LLM FOR ZERO-SHOT SELECTION OF THE OPTIMAL OOD DETECTOR

Section §4.2 discusses the baseline *GPT-4o mini*, where LLM is used as a zero-shot meta-predictor. In this baseline, the prompt provided to the LLM is structured as follows, with text descriptions of both the datasets and models provided. To ensure consistency, we set *temperature* parameter to 0, and *top_p* parameter to 0.999.

```
[Dataset descriptions provided]

 Your task is to select the best OOD detection method for a set
of 24 test ID-OOD dataset pairs.  You will be provided with
descriptions of both the ID-OOD dataset pairs and the available
OOD detection methods.  You should pick the best model that has
the highest AUROC metric.  For each dataset pair, output the
recommended OOD detection method in the format:  'Recommended
Method:  [Recommended Method]'.

 [Model descriptions provided]
```

## B.2 FULL PERFORMANCE RESULTS

Below is the full performance matrix $P$, which shows the performance of eleven OOD detection models (refer to Tab. 1) on our constructed ID-OOD dataset pairs (see dataset pair construction in §4.1). We report AUROC as the metric for OOD detection.

| ID Dataset | OOD Detector | CIFAR-10 | CIFAR-100 | MNIST | Places365 | SVHN | Texture | TIN | SSB_hard | NINCO | iNaturalist | Textures | OpenImage-O |
|---|---|---|---|---|---|---|---|---|---|---|---|---|---|
| CIFAR-10 | Openmax | N/A | **90.68** | 93.95 | **91.17** | **93.31** | **92.71** | **89.67** | 82.53 | 91.53 | 90.07 | 92.85 | 91.36 |
| | MCD | N/A | 88.47 | 92.88 | 88.31 | 90.31 | 87.22 | 86.94 | 80.15 | 88.59 | 85.39 | 87.37 | 87.88 |
| | ODIN | N/A | 65.91 | 88.58 | 67.71 | 94.25 | 72.44 | 63.28 | 58.85 | 65.52 | 50.47 | 72.59 | 62.08 |
| | Mahalanobis | N/A | 82.77 | 94.38 | 83.92 | 95.65 | 96.78 | 82.62 | 76.38 | 85.47 | 86.46 | 96.83 | 90.03 |
| | EnergyBased | N/A | 87.25 | 97.45 | 90.12 | 91.56 | 85.42 | 86.60 | 78.43 | 87.57 | 81.52 | 85.30 | 86.94 |
| | Entropy | N/A | 88.29 | **94.05** | 88.91 | 92.55 | 88.91 | 86.97 | 79.08 | 88.42 | 84.97 | 88.95 | 87.88 |
| | MaxLogit | N/A | 87.29 | 97.27 | 90.04 | 91.58 | 85.55 | 86.59 | 78.44 | 87.58 | 81.66 | 85.44 | 86.97 |
| | KLM | N/A | 83.78 | 90.95 | 84.96 | 88.88 | 85.64 | 83.36 | 74.72 | 84.47 | 76.89 | 85.79 | 83.43 |
| | ViM | N/A | 83.03 | 89.84 | 83.54 | 91.62 | 92.14 | 82.16 | 78.18 | 86.19 | 86.09 | 92.16 | 88.00 |
| | MSP | N/A | 87.95 | 93.27 | 88.45 | 92.09 | 88.58 | 86.59 | 78.85 | 88.05 | 84.78 | 88.62 | 87.52 |
| | KNN | N/A | 88.34 | 92.47 | 89.94 | 92.62 | 91.20 | 87.53 | 80.28 | 88.61 | 84.01 | 91.29 | 89.45 |
| CIFAR-100 | Openmax | **77.40** | N/A | 81.30 | 83.77 | 93.71 | 87.38 | **81.11** | 75.60 | 78.40 | 77.22 | 87.57 | 79.23 |
| | MCD | 76.51 | N/A | 81.78 | 76.40 | 68.19 | 71.16 | 79.50 | 75.68 | 76.89 | 76.59 | 71.03 | 73.83 |
| | ODIN | 59.55 | N/A | 82.72 | 63.01 | 81.51 | 64.27 | 62.73 | 60.57 | 64.48 | 54.65 | 64.31 | 62.14 |
| | Mahalanobis | 66.51 | N/A | 73.35 | 70.13 | 85.21 | **89.84** | 74.04 | 71.46 | 72.87 | 75.63 | 90.20 | 74.73 |
| | EnergyBased | 77.15 | N/A | 93.92 | 78.20 | 72.49 | 76.20 | 80.63 | 75.64 | 78.65 | 74.16 | 76.05 | 74.10 |
| | Entropy | 76.45 | N/A | 86.37 | 77.30 | 71.70 | 74.93 | 80.17 | 75.90 | 77.98 | 77.90 | 74.79 | 75.37 |
| | MaxLogit | 77.32 | N/A | 93.15 | 78.33 | 72.55 | 76.27 | 80.78 | 75.83 | 78.78 | 74.72 | 76.13 | 74.37 |
| | KLM | 74.45 | N/A | 86.02 | 75.79 | 73.86 | 75.41 | 78.63 | 74.85 | 76.73 | 76.89 | 75.26 | 74.38 |
| | ViM | 69.50 | N/A | 86.06 | 71.14 | 81.81 | 88.89 | 77.14 | 73.32 | 77.39 | 75.32 | 89.12 | 72.80 |
| | MSP | 75.10 | N/A | 83.04 | 75.85 | 70.02 | 73.41 | 78.41 | 74.48 | 76.37 | 77.13 | 73.26 | 74.21 |
| | KNN | 71.53 | N/A | 67.42 | 70.96 | 83.29 | 79.73 | 78.38 | 74.96 | 75.20 | 73.38 | 79.99 | 71.44 |
| ImageNet | Openmax | 98.12 | 96.77 | 95.89 | 82.52 | 97.63 | 87.95 | 90.73 | **73.66** | **81.04** | **93.65** | **89.65** | **88.34** |
| | MCD | 82.94 | 85.63 | 89.79 | 79.40 | 97.70 | 80.46 | 79.16 | 72.16 | 79.97 | 88.42 | 82.47 | 84.98 |
| | ODIN | 98.74 | 97.85 | 96.56 | 64.67 | 99.80 | 70.86 | 83.64 | 67.57 | 69.10 | 74.27 | 72.09 | 71.39 |
| | Mahalanobis | 82.33 | 80.45 | 88.25 | 58.25 | 72.91 | 89.95 | 80.72 | 47.91 | 62.07 | 61.18 | 89.78 | 70.54 |
| | EnergyBased | 85.13 | 86.78 | 96.79 | 82.45 | 98.35 | 86.74 | 80.97 | 72.35 | 79.70 | 90.59 | 88.73 | 89.14 |
| | Entropy | 85.72 | 88.14 | 94.38 | 81.43 | 98.64 | 83.23 | 81.46 | 73.07 | 81.80 | 91.03 | 85.34 | 87.78 |
| | MaxLogit | 85.68 | 87.32 | 96.07 | 82.64 | 98.55 | 86.40 | 81.38 | 72.75 | 80.40 | 91.13 | 88.41 | 89.25 |
| | KLM | 87.73 | 88.95 | 91.49 | 78.84 | 97.37 | 82.39 | 82.33 | 68.60 | 79.82 | 89.54 | 84.07 | 85.98 |
| | ViM | 94.34 | 94.26 | 95.89 | 77.52 | 95.10 | 91.82 | 87.21 | 63.83 | 77.53 | 87.23 | 92.51 | 87.05 |
| | MSP | 82.94 | 85.63 | 89.79 | 79.40 | 97.70 | 80.46 | 79.16 | 72.16 | 79.97 | 88.42 | 82.47 | 84.98 |
| | KNN | 71.37 | 71.02 | 69.83 | 61.75 | 64.20 | 60.94 | 70.13 | 55.35 | 63.81 | 56.47 | 61.49 | 60.35 |
| FashionMNIST | Openmax | 92.18 | 90.61 | **93.65** | 91.70 | 91.75 | **92.22** | 90.34 | 91.71 | 91.57 | 94.08 | 92.27 | 90.81 |
| | MCD | 86.03 | 86.88 | 85.65 | 89.17 | 93.85 | 86.77 | 88.43 | 87.74 | 87.83 | 89.06 | 86.59 | 88.67 |
| | ODIN | 57.52 | 59.33 | 55.66 | 61.53 | 89.01 | 60.94 | 59.83 | 60.88 | 61.89 | 47.23 | 61.05 | 57.55 |
| | Mahalanobis | 99.46 | 99.44 | 97.22 | 99.35 | **99.90** | 99.75 | **99.33** | 99.47 | 99.40 | 99.63 | 99.77 | 99.44 |
| | EnergyBased | 91.32 | 92.35 | 79.32 | 94.03 | 98.30 | 91.44 | 93.26 | 93.09 | 92.90 | 93.62 | 91.31 | 93.40 |
| | Entropy | 85.69 | 86.69 | 81.76 | 89.00 | 94.27 | 86.97 | 88.21 | 87.58 | 87.65 | 88.12 | 86.88 | 88.32 |
| | MaxLogit | 91.22 | 92.26 | 79.32 | 93.94 | 98.24 | 91.35 | 93.16 | 92.99 | 92.81 | 93.51 | 91.22 | 93.31 |
| | KLM | 71.28 | 73.85 | 64.34 | 79.02 | 91.84 | 77.16 | 77.49 | 75.26 | 75.87 | 76.58 | 77.12 | 77.78 |
| | ViM | 96.06 | 96.23 | 80.93 | 96.84 | 99.22 | 97.69 | 96.65 | 96.58 | 96.86 | 97.44 | 97.71 | 97.03 |
| | MSP | 84.72 | 85.78 | 80.52 | 88.36 | 93.90 | 86.25 | 87.46 | 86.87 | 86.86 | 87.54 | 86.15 | 87.64 |
| | KNN | **97.71** | **97.70** | 98.38 | **97.85** | 99.45 | 98.12 | 97.72 | 97.72 | 97.74 | 97.76 | 98.14 | 97.95 |

Table C: Various OOD detection models' performance on ID-OOD dataset pairs. See experiment setting in §4.1. We highlight the selected OOD method for each dataset on the test set in bold.

### B.3 DATASET DESCRIPTIONS

Below is the full list of the dataset descriptions we use in the study, which includes basic dataset metadata such as the types of objects in each dataset, the image type, and the size of the dataset. This information can be compiled swiftly and easily.

| Dataset | Description |
|---|---|
| CIFAR-10 | Contains images of 10 types of objects, including airplanes, cars, birds, cats, deer, dogs, frogs, horses, ships, and trucks. Each image is a small 32x32 RGB image. |
| CIFAR-100 | Similar to CIFAR-10 but includes 100 classes grouped into 20 superclasses. Each class contains images of specific objects, animals, or people. |
| MNIST | Comprises black-and-white images of handwritten digits from 0 to 9, with each image normalized to fit in a 28x28 pixel bounding box. |
| Places365 | Features images of various places like homes, parks, offices, and cities, spread across 365 different scene categories. |
| SVHN | Includes images of digit sequences found in natural scene images, primarily focusing on house numbers from street view images. |
| Texture | Generally consists of images depicting various surface textures such as bricks, tiles, fabrics, and other patterned surfaces. |
| TIN | Contains tiny 32x32 images sourced from the internet, depicting a wide variety of everyday objects, animals, and scenes. |
| SSB-Hard | Contains images of specific breeds of birds, types of aircraft, and models of cars. |
| NINCO | Contains natural scenes, textures, and objects, providing a diverse set of images. |
| iNaturalist | Consists of images submitted by users of various species of plants, animals, and fungi, used for species identification and classification. |
| Textures | Focuses on images that capture the surface quality of various materials. |
| OpenImages | Features a wide range of images with various objects, scenes, and activities, annotated with labels and bounding boxes. |
| ImageNet | Contains diverse images across a wide range of categories like different types of animals, plants, vehicles, and everyday objects. |
| Fashion-MNIST | Includes images of fashion products from 10 categories, such as shirts, dresses, shoes, and bags. Each image is a 28x28 grayscale image. |

Table D: Dataset descriptions used in the study, which contain basic information of the dataset such as dataset content (e.g., what kind of objects are in the dataset), image type, and dataset size.

### B.4 FEATURE IMPORTANCE OF LANGUAGE EMBEDDINGS

Fig. A shows the feature importance of the language embeddings, where the F score of a feature is the total number of times the feature is used to split the data in all trees in the model. We show the top-10 most important features. According to the analysis, the initial dimensions of these embeddings carry greater significance.

| Embeddings / Metric | ID dataset | OOD dataset | Method |
|---|---|---|---|
| Weight | 0.53 | 0.52 | 0.37 |
| Cover | 7.27e-07 | 2.88e-07 | 7.08e-07 |
| Gain | 0.026 | 0.025 | 0.018 |

Table E: Average feature importance for different language embeddings on the Weight, Cover, and Gain metrics. Dataset embeddings play a more significant role in the selection process compared to method embeddings.

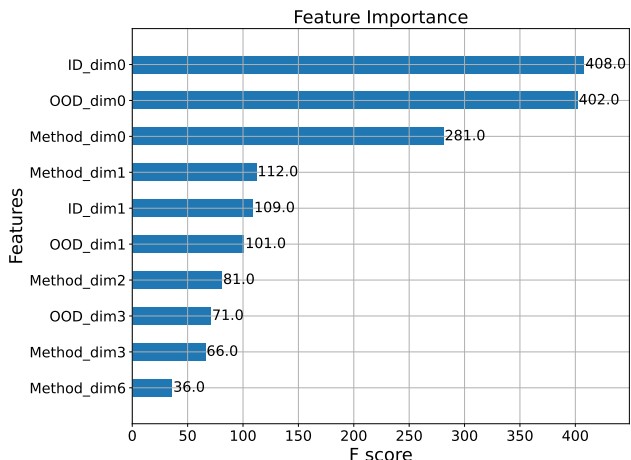

Figure A: Top-10 feature importance of the language embeddings used in the study; dimensions from the ID dataset are labeled as ID_dim, those from the OOD dataset as OOD_dim, and from the Method as Method_dim. The initial dimensions of language embeddings are the most critical.

Tab. E presents the average feature importance based on three metrics: 1) Weight: the number of times a feature is used to split the data across all trees, 2) Cover: the average coverage across all splits the feature is used in, and 3) Gain: the average gain across all splits the feature is used in the ID dataset embeddings. These metrics are computed for the ID dataset embeddings, OOD dataset embeddings, and method embeddings. The results indicate that dataset embeddings have a greater impact on the selection process compared to method embeddings.

## C  FURTHER STUDY ON EMBEDDINGS

### C.1  COMBINED META-FEATURE AND LANGUAGE EMBEDDING

Some might argue that relying solely on language embeddings could be overly basic. To address this, we also experimented with combining statistical and landmarker meta-features alongside language embeddings for dataset and method embeddings (denoted as Combined). As shown in Fig. B, this approach achieves performance comparable to MetaOOD. However, the computational and time costs of extracting statistical features can be substantial, particularly for large-scale datasets like ImageNet and LSUN. Language embeddings were chosen primarily for their reproducibility, efficiency, and significantly lower computational overhead.

### C.2  VARIATION OF DATASET DESCRIPTIONS

To test the robustness of language embeddings against different input descriptions of datasets, we manually adjust dataset descriptions by rephrasing them to alter the wording while preserving the original meaning, as detailed in Tab. F. This variation is referred to as MetaOOD'. According to Fig. C, changes in dataset descriptions containing basic information do not lead to significant differences, and performance remains stable.

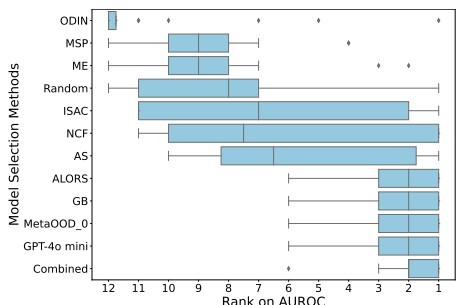

Figure B: Boxplot of the rank distribution of combined meta-feature and language embedding, and baselines. The performance is comparable to that of MetaOOD.

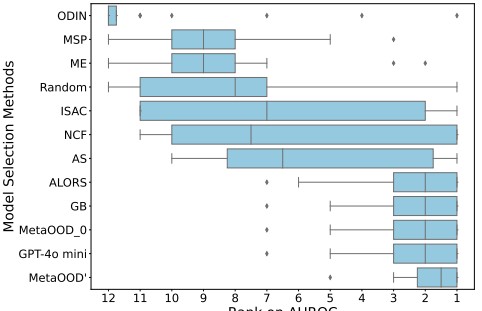

Figure C: Boxplot of the rank distribution of variation of dataset descriptions and baselines. Variations in dataset descriptions, which include basic information, do not lead to significant differences, and performance remains stable.

| Dataset | Description |
|---------|-------------|
| CIFAR-10 | A standard dataset in computer vision, containing 60,000 color images (32x32 pixels) of 10 distinct object categories: airplanes, cars, birds, cats, deer, dogs, frogs, horses, ships, and trucks. |
| CIFAR-100 | An extension of CIFAR-10, featuring 100 classes grouped into 20 broader superclasses. It includes 60,000 color images (32x32 pixels), each class representing a finer granularity of objects, animals, or people. |
| MNIST | A seminal dataset comprising 70,000 grayscale images (28x28 pixels) of handwritten digits (0-9), each centered in a bounding box. |
| Places365 | Contains 1.8 million images spanning 365 distinct scene categories, including various indoor and outdoor environments like homes, parks, offices, and cities. |
| SVHN | A real-world image dataset featuring over 600,000 images of digit sequences (32x32 pixels) extracted from house numbers in street view imagery. |
| Texture | Composed of diverse images highlighting surface textures such as bricks, tiles, fabrics, and other patterned materials, often used to study material properties. |
| TiN | A massive dataset of 80 million tiny images (32x32 pixels) collected from the web, representing a broad spectrum of everyday objects, animals, and scenes. |
| SSB Hard | Features a diverse collection of images, including specific bird species, distinct types of aircraft, and detailed models of cars. |
| NINCO | Comprises a wide range of images, including natural scenes, intricate textures, and various objects. |
| iNaturalist | Comprises over 859,000 images of various species of plants, animals, and fungi, contributed by users for species identification and classification. |
| Textures | Similar to the Texture dataset, but may have a different focus or a wider range of surface textures, including natural and artificial materials. |
| OpenImages | A vast dataset containing 9 million images annotated with labels, bounding boxes, and object segmentation across diverse objects, scenes, and activities. |
| ImageNet | One of the most comprehensive datasets, with over 14 million images categorized into 1,000 classes, including animals, plants, vehicles, and various everyday objects. |
| Fashion-MNIST | A modern alternative to MNIST, containing 70,000 grayscale images (28x28 pixels) of fashion items categorized into 10 classes, such as shirts, dresses, shoes, and bags. |

Table F: Manually modified dataset descriptions used in additional study C.2, which contain basic information of the dataset such as dataset content (e.g., what kind of objects are in the dataset), image type, and dataset size.

