# OpenReview forum: "MetaOOD: Automatic Selection of OOD Detection Models"
_ICLR.cc/2025/Conference — ICLR 2025 Poster_

### Official Review · Reviewer_kAtw · 2024-10-30

**Soundness:** 3
**Presentation:** 2
**Contribution:** 3
**Rating:** 6
**Confidence:** 5

**Summary:**

The paper presents MetaOOD, a framework for automatic selection of out-of-distribution (OOD) detection models without requiring labeled data. It leverages historical performance data and language model embeddings. The approach aims to improve the reliability of OOD detection in critical applications, such as autonomous driving and online transactions. Overall, MetaOOD addresses the challenge of adapting to data shifts effectively and efficiently.

**Strengths:**

The paper's strengths include the introduction of a zero-shot, unsupervised framework for OOD detection model selection, which enhances adaptability to new datasets. It effectively utilizes language model-generated embeddings to capture nuanced dataset characteristics, improving model selection accuracy. The extensive experimentation demonstrates superior performance compared to eleven established methods, showcasing its robustness. Additionally, the framework incurs minimal runtime overhead, making it efficient for practical applications. The use of the Wilcoxon signed-rank test is a plus of the paper. The p-values suggests the proposed approach works well.

**Weaknesses:**

The paper's weaknesses include a reliance on the quality of language model embeddings, which may vary based on the model used and the nature of the input data. Additionally, the framework's performance may be limited by the diversity of the historical data pool, potentially affecting generalization to unseen datasets. The lack of extensive real-world testing could raise concerns about its applicability in practical scenarios. Lastly, the complexity of the approach may pose challenges for reproducibility and implementation in different contexts. Obtain datasets and model feature/embeddings from their textual descriptions appear a bit strange and somewhat unreliable.

**Questions:**

No questions.

---

> ### Author Response · Authors · 2024-11-20
>
> > W1. The paper's weaknesses include a reliance on the quality of language model embeddings, which may vary based on the model used and the nature of the input data.
>
> Thank you for your thoughtful feedback. To mitigate this concern, we have utilized popular and widely adopted language models such as HuggingFace BERT-based models, OpenAI embedding model, and LLaMA. Our experiments demonstrate that our approach maintains strong performance across these models, highlighting its robustness.
>
> > W2. Additionally, the framework's performance may be limited by the diversity of the historical data pool, potentially affecting generalization to unseen datasets. The lack of extensive real-world testing could raise concerns about its applicability in practical scenarios.
>
> This is so true: as a meta-learning algorithm, MetaOOD also depends on the similarity between the task to the meta-train/historical datasets. To improve the generalization to unseen datasets, we have incorporated widely-used OOD detection benchmark datasets and leveraged the PyTorch-OOD library, which offers a unified interface for implementing OOD detection methods. This integration ensures that our framework is adaptable and can be further applied to additional datasets and OOD detection methods.
>
> > W3. Lastly, the complexity of the approach may pose challenges for reproducibility and implementation in different contexts. Obtain datasets and model feature/embeddings from their textual descriptions appear a bit strange and somewhat unreliable.
>
> We also recognize the importance of reproducibility and ease of implementation. To address potential complexities, we have made our code publicly available. We hope this could allow other researchers and users to replicate our results and implement our approach in different contexts with minimal difficulty. The dataset description contains basic information such as dataset content (e.g., what kind of objects are in the dataset), image type, and dataset size, as shown in the example, which can be extended to unseen datasets easily and quickly. We added the full list of dataset descriptions in appendix Table D.
>
> Again, we appreciate your insights and believe that our efforts in these areas help address the concerns raised, contributing to the novelty and practical applicability of our work.

---

> > ### Comment · Reviewer_kAtw · 2024-11-26
> > **Thanks for the comments.**
> >
> > Thanks for the comment.
> >
> > I will keep my score.

---

### Official Review · Reviewer_N69E · 2024-11-02

**Soundness:** 2
**Presentation:** 2
**Contribution:** 3
**Rating:** 6
**Confidence:** 5

**Summary:**

The paper postulates that by identifying which OOD detection models have historically performed well on datasets similar to the one currently being considered, one can select the model most likely to be effective without needing labels for supervised training.  A meta-learning approach is used to take past performance data from various models (across data sets); when new dataset arrives, the approach checks for similarity between the new dataset and historical ones using embeddings. The assumption is that the selected model will perform well as it is closest to the data set under use.  Meta-learning (training) is done offline with curated data sets; while OOD model selection is online as a specific data point arrives.  Results show that their approach works better than compared other techniques.  Experiment approach itself is reasonable and approach is sound.

**Strengths:**

Integration of language model based embeddings; Empirical eval. is done to good detail. this technique is actually useful. though it is a logical next step ~ the approach itself can be used in other contexts or at-least the idea can be adapted. Sufficient detail is provided makes work transparent.

**Weaknesses:**

Eval is too narrow and limited ~ so results may not generalise or this approach may be limited to the data set / domain attempted; esp. since there is no formal conceptual development as such we do not know when and where this method will work or have an intuition for where the limit may be.  Although there is the claim of unsupervised world-first etc. ~ there is still a need for other forms of supervised and curated training. Assumes text descriptions are good in the evals and curated data sets - but does it hold for the real world?  All the usual limits of language models apply here too. Scalability not known -- again since we do not have specific underlying theory.

**Questions:**

Can the authors add a diagram to better illustrate how this technique would work in practice? This will help translatability of this work into other contexts faster.

---

> ### Author Response · Authors · 2024-11-20
>
> > W1. Eval is too narrow and limited ~ so results may not generalise or this approach may be limited to the data set / domain attempted; esp. since there is no formal conceptual development as such we do not know when and where this method will work or have an intuition for where the limit may be.
>
> Thank you for your comments. For the dataset/ domain issue, we tried to mitigate its effect by incorporating well-known OOD detection benchmark datasets (CIFAR-10, CIFAR-100 and ImageNet). We recognize that OOD detection benchmarking can be constrained by computational demands due to the large size of datasets (particularly ImageNet) and the limited availability of standard, diverse OOD data, which may affect generalizability across other domains.
> As a meta-learning method, MetaOOD has the capacity to generalize to the case (better than random selection) even when the similarity to the meta-train datasets is weak. Meanwhile, this is also its limitation that the similarity to the meta-train datasets is assumed to be.
> Following your suggestion, we revised our text to reflect this limitation and highlight a few future directions: (i) add uncertainty quantification to the prediction so it can say “I do not know” and (ii) default the selection to the global best OOD detector on the meta-train when the uncertainty is high (the prediction confidence is low).
>
> > W2. Although there is the claim of unsupervised world-first etc. ~ there is still a need for other forms of supervised and curated training.
>
> Thank you for highlighting the importance of various training methods. This task is designed for an unsupervised OOD detection setting, as we can extend the proposed framework to additional methods and datasets with minimal effort. Note the goal of this work is not to rule out semi/supervised methods, but to offer a fast, zero-shot approach for selection.
> Indeed, the training of the meta-regressor is supervised on the meta-train datasets, which offers us great capacity in selection. So we can clarify that MetaOOD is designed for unsupervised OOD methods (able to extend to semi/supervised one), while the training of the meta predictor is supervised on the meta-train datasets.
> We will clarify this better in our paper, and thank you for pointing out this to make it clearer paper!
>
> > W3. Assumes text descriptions are good in the evals and curated data sets - but does it hold for the real world? All the usual limits of language models apply here too. Scalability not known -- again since we do not have specific underlying theory.
>
> Thank you for your valuable feedback. Using text descriptions in evaluations and curated datasets has limitations, and real-world applicability may vary. In this study, we focused on testing with several well-known language models (e.g., Hugging Face, OpenAI embedding model, and LLaMA) to provide preliminary insights into generalizability and practical performance across various scenarios. The first few dimensions of the method and dataset embeddings matter more based on our feature importance analysis (we added Figure B in appendix for illustration).
>
> > Q1. Can the authors add a diagram to better illustrate how this technique would work in practice? This will help translatability of this work into other contexts faster.
>
> Thank you for suggesting the inclusion of an additional diagram for better illustration. We have added a detailed overview of the MetaOOD method in Appendix Figure A, along with a description of the notations used in the figure in Appendix Section A.2.

---

> > ### Comment · Reviewer_N69E · 2024-11-27
> >
> > The feedback from the authors across the various reviewers has been helpful for clarification. I have been supportive of this work & remain the same now as well.

---

### Official Review · Reviewer_bKXx · 2024-11-02

**Soundness:** 1
**Presentation:** 1
**Contribution:** 1
**Rating:** 3
**Confidence:** 4

**Summary:**

This paper presents MetaOOD, a “model” selection approach for out-of-distribution (OOD) detection. MetaOOD utilizes language models to generate feature embeddings of both the meta dataset and “models”, allowing for the optimal “model” selection based on anticipated performance on the test set. The results on the Wilcoxon statistical tests show the promising performance of MetaOOD.

**Strengths:**

1. The motivation is sound. It is interesting to see a meta-selection approach to the OOD detection problem since there are so many methods in this OOD domain.
2. The proposed method is simple and straightforward. The results on the traditional methods are promising.

**Weaknesses:**

1. The definition of "OOD model" is confusing. There are many post-hoc detection methods in the detection problem, which should not be classified as “models”. For instance, the paper includes the MSP method for the selection experiments. However, MSP is just a simple post-hoc technique that can be applied to most classification models (e.g., ResNet) using the SoftMax function. This method should not be considered as a model, which is misleading considering another factor, “model architecture,” in the experiments.

2. The methodology lacks depth. MetaOOD merely utilizes language models to extract embeddings for dataset and model descriptions, and then select the top-1 method based on these embeddings. The approach lacks insight and overlooks potential issues. For instance, the embeddings derived from descriptions may not accurately capture the true characteristics of the models and datasets. Also, simply selecting the top 1 can overlook the nuances of methods and the potential problems of the utilized datasets.

3. The experimental results are unconvincing. The baseline methods included are outdated, with the most recent method (NCF) dating back to 2017.

4. The terms OOD and OOD detection should not be used interchangeably. It is unclear what is meant by "OOD dataset" given such a name strategy. Is it referring to a commonly recognized OOD dataset distinct from the in-distribution (InD) dataset, or simply an OOD detection dataset (includes train, val, and test splits for detection methods)?

5. I am curious whether this paper was generated by a language model, such as GPT-4. The writing style, particularly in Section 3.3.1, resembles AI-generated text. Given the simplicity of the method, the Method Section could be more concise, potentially requiring only 0.5 pages to convey the core elements of the approach. However, the current version spans 2.5 pages.

**Post-rebuttal Comments**

I'd like to thank the authors' response. I agree that weakness 5 could be too strong as I assumed the method may be generated from GPTs. However, the length and the redundancy of the method section are still a problem. I still cannot accept that the choice of embedding strategy could occupy half of the Method Section, given that the embedding method is that simple.

I also agree that the simplicity could be the advantage of the proposed method. However, I still cannot view this method as sufficiently advanced or novel for a research paper at ICLR. As I've mentioned, this method lacks depth, not to mention the absent theoretical insights. The approach merely utilizes the rough descriptions of the dataset and models to predict the score on the OOD dataset. The performance relies heavily on LLMs, which can introduce a series of problems and are heavily limited to the application scenarios.

Also, I acknowledge that the paper may present the "first" selection strategy for the OOD detection method. However, this should not be an excuse for the obvious shortcomings. Overall, given current experiments, I consider this paper more of an exploration of leveraging LLMs in OOD detection rather than a substantial research contribution. Thus, I will maintain my original score.

**Additional Post-rebuttal Comments**

After reviewing the authors' responses, I still consider this paper as a preliminary exploration of leveraging LLMs in OOD detection. So I will maintain my original score.

**Questions:**

NA

---

> ### Author Response · Authors · 2024-11-20
>
> > Q1. The definition of "OOD model" is confusing. There are many post-hoc detection methods in the detection problem, which should not be classified as “models”. For instance, the paper includes the MSP method for the selection experiments. However, MSP is just a simple post-hoc technique that can be applied to most classification models (e.g., ResNet) using the SoftMax function. This method should not be considered as a model, which is misleading considering another factor, “model architecture,” in the experiments.
>
> Thank you for pointing this out. We also believe that referring post-hoc OOD detection methods like MSP may be better as “method”. Calling them models is more for following the tradition of model selection research. We added an explanation in footnote of page 3 (Section 3.2) for clarification.
>
> > Q2, The methodology lacks depth. MetaOOD merely utilizes language models to extract embeddings for dataset and model descriptions, and then select the top-1 method based on these embeddings. The approach lacks insight and overlooks potential issues. For instance, the embeddings derived from descriptions may not accurately capture the true characteristics of the models and datasets. Also, simply selecting the top 1 can overlook the nuances of methods and the potential problems of the utilized datasets.
>
> We believe a simple, effective approach is always preferred in research. Thus, we respectfully point out that many of the leading research in model selection are rather simple. MetaOD [NeurIPS 2021], MetaGL [ICLR 2023] and AdGym [NeurIPS 2023] use meta-learning approach and models that link embedded data representations to performance outcomes.
> The key contribution of this work is to (1) 1st OOD method selection (2) a novel way for embedding dataset and model.
> In the real world, users would mostly care about the top-1 model as only one model usage may be preferred. We thus focus on the top-1 model selection in our study. However, it can be extended to top-k model selection as the model selection approach would generate predicted performance for all the available OOD detection methods. In our research, we compared the traditional statistical and landmarker meta-features with language embeddings, finding that language embeddings provide a faster and more effective solution. While we acknowledge that language embeddings may not capture every nuanced characteristic, we aimed to demonstrate a quick and reliable approach for selecting OOD detection methods. Further research could deepen the exploration of language embeddings to address more intricate aspects.
>
> > Q3. The experimental results are unconvincing. The baseline methods included are outdated, with the most recent method (NCF) dating back to 2017.
>
> As the first of its kind for OOD detection, we do not have an immediate baseline. Thus, we follow the tradition to compare based on unsupervised model selection methods [1].
> As supervised methods do not apply to our task with ground truth labels unavailable for OOD detection, we look at existing unsupervised methods for our task. We also consider zero-shot LLM (GPT-4o) as method selector as one of our baseline which is more recent (2024). Below is the result with comparison to method used in [1] (2021). MetaOOD also demonstrates better and more stable performance (p-value<0.05 and lower average rank). To the best of our knowledge, we have made an effort to consider the existing methods for comparison.
>
> |   Wilcoxon-test      | MetaOD & MetaOOD   |
> |---------|--------------------|
> | p-value |        0.0064            |
>
> |         | MetaOD | MetaOOD |
> |---------|--------|---------|
> | avg rank|   6.5     |   1.583      |
>
>
> [1] Zhao, Y., Rossi, R., & Akoglu, L. (2021). Automatic unsupervised outlier model selection. Advances in Neural Information Processing Systems, 34, 4489-4502.
>
> > Q4, The terms OOD and OOD detection should not be used interchangeably. It is unclear what is meant by "OOD dataset" given such a name strategy. Is it referring to a commonly recognized OOD dataset distinct from the in-distribution (InD) dataset, or simply an OOD detection dataset (includes train, val, and test splits for detection methods)?
>
> Thank you for your feedback. As we had in the first page footnote – we may omit detection to save space. we have correctted all the corresponding OOD terms and marked the changes in blue in the updated file.

---

> ### Author Response · Authors · 2024-11-20
>
> > Q5. I am curious whether this paper was generated by a language model, such as GPT-4. The writing style, particularly in Section 3.3.1, resembles AI-generated text. Given the simplicity of the method, the Method Section could be more concise, potentially requiring only 0.5 pages to convey the core elements of the approach. However, the current version spans 2.5 pages.
>
> First, this paper uses LLM only for fixing grammar and re-wording sentences as disclosed.
> Second, we believe the presentation should be in details for general audience. We appreciate and are impressed by your finding our method simple and clear, while that may be contributed to your expertise in the field.

---

> ### Author Response · Authors · 2024-11-27
>
> Thank you for your updated feedback. We greatly appreciate your acknowledgment of the **simplicity of our method as an advantage** and its distinction as the **first** unsupervised approach for OOD detection method selection. However, we think there may be some misunderstandings. There are two types of embeddings we discuss and compare in the study: 1) the statistical +landmarker meta-feature embedding and 2) language embedding. Although we find language embedding is better (in terms of reproducibility, speed, and computational cost), the traditional meta-feature method is also crafted and studied in this research. We thus include descriptions of both features in the method section (the complete list of the statistical+landmarker feature and language embedding input are in the appendix Table B and Table D, respectively). Therefore, the space spans half the method section. It is not merely a matter of selecting an embedding; rather, our choice follows a thorough comparison with traditional statistical features that we carefully crafted. Our experiments demonstrate that our approach maintains strong performance across these models, highlighting its robustness. This research also provides insight to the use of different embeddings within the meta-learning framework. Within language embeddings, we also conduct ablation study on popular and widely adopted language models such as HuggingFace BERT-based models, OpenAI embedding model, and LLaMA (Figure 4).
> If using both statistical embedding and language embedding, the performance would be comparable.
> |       | p-val (compared to MetaOOD) | avg_rank |
> |-------|-------|----------|
> | **Combined feature** |  0.0687    | 1.875        |
>
> However, the time and computational cost of statistical feature can be huge especially when dealing with large datasets such as Imagenet, LSUN... We have also made the code for generating the statistical available. The primary reasons for selecting language embeddings are their reproducibility, efficiency, and lower computational requirements. The requirement of the basic information of the dataset guarantees the generalizibility of the framework. Moreover, the meta-learning methodology, which makes use of the similarity of the historical task to the target task, serves as the foundation that enables the approach to function effectively [1].
>
> For soundness and interpretation, we find that the first few dimensions of the method and dataset embeddings matter more based on our feature importance analysis (we added Figure B in appendix for illustration). Also, according to appendix Table E, dataset embeddings have a greater impact on the selection process compared to method embeddings. Further interpretability of language embeddings, as suggested in existing literature in the NLP field, remains an area for future study. One direction may be using language models for explanation.
> We also add an experiment on dataset descriptions that were manually varied (refer to as MetaOOD'):
> |       | p-val (compared to MetaOOD) | avg_rank |
> |-------|-------|----------|
> | **MetaOOD'** |  0.1763    | 1.833       |
>
> The p-value shows that variations in dataset description which includes basic information, do not lead to significant differences, and performance remains stable. We will include the boxplot figures for both the experiments discussed above (combined feature and MetaOOD') into the appendix as well. Thank you!
>
> [1]Chelsea Finn, Pieter Abbeel, and Sergey Levine. 2017. Model-agnostic meta-learning for fast adaptation of deep networks. In Proceedings of the 34th International Conference on Machine Learning - Volume 70 (ICML'17). JMLR.org, 1126–1135.

---

### Official Review · Reviewer_nA6W · 2024-11-03

**Soundness:** 3
**Presentation:** 3
**Contribution:** 3
**Rating:** 6
**Confidence:** 4

**Summary:**

In this paper, the authors propose MetaOOD, which utilizes meta-learning to select an OOD detection model automatically. The motivation is that each OOD detection algorithm might excel in specific scenarios but may not perform well universally, therefore it is important to select one particular OOD detection for each task. MetaOOD utilizes historical performance data of existing methods across a variety of benchmark out-of-distribution (OOD) datasets to enable efficient model selection for new datasets, eliminating the need for labeled data at test time. To more accurately measure task similarities, the authors incorporate language model-based embeddings that capture the unique OOD characteristics of both datasets and detection models. Through extensive testing across 24 unique test dataset pairs and 11 OOD detection models, the authors show that MetaOOD consistently outperforms current methods with minimal additional computation time.

**Strengths:**

1. The idea of using meta-learning to select the best OOD detection method for each specific task is interesting.
2. The paper is generally easy to understand and clearly written.
3. The experiments show the effectiveness of the proposed method.

**Weaknesses:**

1. Figure 1 needs to be improved. The notations in the figure are confusing and unclear.
2. The design of the textual description seems ad-hoc and cannot be applied in the case of without detailed dataset information.
3. Detailed results on the selected OOD method for each dataset are missing.

**Questions:**

1. Does the proposed method rely on the architecture of the trained model?
2. What is the training time of the proposed method?
3. If there is one additional OOD method, how can incorporate this method into the proposed MetaOOD?
4. What are the main factors that influence the choice of an OOD method based on the characteristics of the training and test sets?

---

> ### Author Response · Authors · 2024-11-20
>
> > W1. Figure 1 needs to be improved. The notations in the figure are confusing and unclear.
>
> Thanks for the suggestion on the flowchart figure. In this revision, we have added text descriptions of notations for embeddings, meta-predictor and predicted performance to Figure 1. Also, we added a comprehensive overview of the MetaOOD method in appendix figure A, with the notations used in the figure listed in appendix section A.2.
>
> > W2. The design of the textual description seems ad-hoc and cannot be applied in the case of without detailed dataset information.
>
>
> The dataset description contains basic information such as dataset content (e.g., what kind of objects are in the dataset), image type, and dataset size, as shown in the example, which can be extended to unseen datasets easily and quickly.
> For instance, for an additional dataset pair like ImageNet-LSUN, we would generate the language embeddings of the Imagenet-LSUN dataset pair based on their language descriptions such as:
> - For Imagenet: Contains diverse images across a wide range of categories like different types of animals, plants, vehicles, and everyday objects.
> - For LSUN: Contains high-resolution images across various scene categories such as bedrooms, living rooms, churches, and outdoor spaces, as well as specific objects
>
> and then perform the model selection on our trained model.
>
> > W3. Detailed results on the selected OOD method for each dataset are missing.
>
> Thank you for pointing that out. We highlighted the selected models in Table B per dataset in the appendix.
>
> > Q1. Does the proposed method rely on the architecture of the trained model?
>
> The proposed method is agnostic to the trained model as MetaOOD is a framework. The architecture of the model trained on the in-distribution (ID) data can impact the performance of certain OOD detection methods that require fitting to the data, which may impact the performance metric and further but not impact the proposed model selection method.
> On the model selection part, we chose the XGBoost tree model for our model selection method (model) because it is both fast and demonstrates stable performance, as used in similar research. We also experimented with a neural network structure and found it is consistent with the finding [1] that the XGboost tree structure offers both stability and superior performance.
> To sum up, the trained ID model may affect OOD performance, while MetaOOD is agnostic to this.
> [1] Jiang, M., Hou, C., Zheng, A., Han, S., Huang, H., Wen, Q., ... & Zhao, Y. (2024). Adgym: Design choices for deep anomaly detection. Advances in Neural Information Processing Systems, 36.
> > Q2. What is the training time of the proposed method?
>
> | Training Time (s) | Dataset Embedding Generation (s) | Method Embedding Generation (s) |
> |--------------------|--------------------------------------|----------------------------------|
> | 89.1             | 5.1                                  | 3.5                              |
>
> The training time of the proposed method, including the offline data and method embedding generation phase and the selection model training phase, can be done within dozens of seconds. The embedding generation is efficient with the use of LLM, and the training of the OOD method selection model is quick, given the structure and stability of the XGBoost tree model.
>
> > Q3. If there is one additional OOD method, how can incorporate this method into the proposed MetaOOD?
>
> Additional OOD detection methods can be added to the performance matrix and incorporated into this unsupervised model selection approach using meta-learning. Since the training of the model selection model is fast, once the performance result of the additional OOD detection method is available, one can easily add the information to the performance matrix and train the model selection model (we make the training code available) with ease. Notably, our current performance matrix is run based on the publicly available pytorch-ood library, which can be expanded with additional methods as well.
>
> > Q4. What are the main factors that influence the choice of an OOD method based on the characteristics of the training and test sets?
>
> The similarity between the training and test sets would be one contributing factor to this meta-learning-based approach. The meta-learning approach leverages previous learning experiences to learn general patterns to expand the model’s ability to adapt to different scenarios.
> We investigated the feature importance of the embeddings and found that the initial dimensions of the language embeddings play a more significant role in model selection (we added Figure B in the appendix for illustration). Further interpretability of language embeddings, as suggested in existing literature in the NLP field, remains an area for future study. One direction may be using language models for explanation.

---

> > ### Comment · Area_Chair_TcCL · 2024-12-01
> >
> > Dear Reviewer nA6W,
> >
> > The authors have provided responses - do have a look and engage with them in a discussion to clarify any remaining issues as the discussion period is coming to a close in less than a day (2nd Dec AoE for reviewer responses).
> >
> > Thanks for your service to ICLR 2025.
> >
> > Best,
> > AC

---

> ### Comment · Reviewer_nA6W · 2024-12-02
>
> The rebuttal addresses my concerns and I would like to maintain my original rating. I would encourage the authors to include the additional results into the final version of the paper.

---

### Author Response · Authors · 2024-11-20

### Summary of Our Responses and Contributions
We sincerely thank all reviewers for their thoughtful and constructive feedback, as well as for recognizing the novelty and importance of our work. We have carefully addressed each individual comment and made revisions to improve the manuscript (changes highlighted in blue). Below, we summarize the major contributions of our work and highlight points endorsed by the reviewers:

**Novelty**: We introduced **MetaOOD**, the first zero-shot, unsupervised framework for automatic selection of OOD detection models. We are grateful that multiple reviewers (e.g., Reviewer N69E and kAtw) acknowledged the significance of this problem, describing it as a "logical next step" and an "effective and efficient" solution to the critical challenge of adapting OOD detection to real-world data shifts in domains such as autonomous driving and healthcare.

**Specialized Framework**: MetaOOD leverages **meta-learning** with language model-based embeddings to capture the distinctive characteristics of datasets and OOD detection models. This enables robust model selection without requiring labeled data. Reviewers kAtw and nA6W highlighted using embeddings and meta-learning as a **"sound and interesting"** approach that offers a principled solution to an underexplored challenge.

**Extensive Experimental Validation**: Our experiments demonstrate the effectiveness and efficiency of MetaOOD across 24 test dataset pairs and 11 OOD detection models, significantly outperforming existing methods with minimal computational overhead. Reviewers kAtw and N69E appreciated the robustness of our results, with Reviewer kAtw commending our use of the **Wilcoxon statistical test** to validate performance claims. Reviewer nA6W noted that the experimental setup is "extensive" and the results convincingly support our claims.

**Practical Contributions**: MetaOOD enhances reliability in critical open-world applications by automating model selection, and eliminating the need for manual tuning or labeled data. Reviewer N69E highlighted its potential for broader applicability, stating that "the approach itself can be adapted to other contexts" and offers valuable contributions to practical OOD detection.

---

> ### Author Response · Authors · 2024-12-02
>
> We would like to take this opportunity to briefly summarize the additions made to the appendix during the rebuttal phase, which we would also like to include in the final version of the paper. These include Figure A and its associated notations in Section A.2, and a comprehensive dataset description in Table D of the appendix. Furthermore, we expanded our analysis on language embeddings in Sections B.4 and C. This includes: (1) assessing the feature importance of language embedding, (2) introducing a variant of MetaOOD that incorporates combined statistical and landmarker meta-features alongside language embeddings, and (3) examining the impact of various dataset descriptions. Additionally, we observed that recent studies have also highlighted the advantages of using LLM embeddings over traditional feature engineering for high-dimensional regression tasks [1].
>
> [1] Tang, E., Yang, B., & Song, X. (2024). Understanding LLM Embeddings for Regression. arXiv preprint arXiv:2411.14708.

---

### Author Response · Authors · 2024-11-25

We sincerely thank the reviewer again for your valuable time and thoughtful comments. We have updated the paper in response to your feedback, with changes highlighted in blue in the updated file and additional tables/figures included in the appendix. As we are approaching the end of the discussion stage, we would greatly appreciate it if you could kindly read our responses and update the scores if your concerns have been addressed. We are more than happy to further discuss any concerns that you find not fully addressed. Thank you very much.

---

### Meta-Review · Area_Chair_TcCL · 2024-12-24

**Metareview:**

This paper presents a meta-learning framework for model selection of out-of-distribution detection models. The core idea is to use embeddings from language models to represent datasets and models as described by textual descriptions. The framework is evaluated on image-based OOD detection tasks and shown to outperform other baselines.

The paper addresses an important practical problem of model selection for OOD detection. The proposed method is simple and shown to be effective, using rigorous statistical tests. The work could be strengthened by additional analysis on the embeddings and how they are capturing task and dataset similarity, as performance appears to be somewhat sensitive to choice of language model used (Fig 4). Experiments on a broader range of datasets will also help to strengthen evidence for the generalizability of the approach.

Overall, the AC leans towards accepting this paper as an interesting first approach to an important practical problem. The authors should incorporate all changes and additional results as promised in the discussion.

**Additional Comments On Reviewer Discussion:**

Most reviewers had concerns regarding the generalizability of the approach, which the authors addressed to most reviewers' satisfaction through explanations and updates to the text. One reviewer had concerns about the limited analysis of the method and lack of comparison to more recent baselines, which the AC thinks was somewhat addressed with additional results, though the reviewer remained unconvinced at the end of the discussion. Overall, the AC agrees that more analysis of the method should be provided, but leans positive due to the promising results and simple, novel approach.

---

### Decision · Program_Chairs · 2025-01-22

Accept (Poster)